# Frailty, a Dimension of Impaired Functional Status in Advanced COPD: Utility and Clinical Applicability

**DOI:** 10.3390/medicina57050474

**Published:** 2021-05-11

**Authors:** Sabina Antonela Antoniu, Lucian Vasile Boiculese, Virgiliu Prunoiu

**Affiliations:** 1Department of Medicine II and L Boiculese, Department of Interdisciplinary Medicine, University of Medicine and Pharmacy Grigore T Popa, 700115 Iasi, Romania; lboiculese@gmail.com; 2Department 10 Surgery, University of Medicine and Pharmacy Carol Davila, 020021 Bucharest, Romania; virgiliuprunoiu@yahoo.com

**Keywords:** aging-COPD-frequent exacerbators, functional status, mortality, physical frailty phenotype, prognosis

## Abstract

*Background and Objectives*: In advanced chronic obstructive pulmonary disease (COPD), functional status is significantly impaired mainly as a result of disease related respiratory symptoms such as dyspnea or as a result of fatigue, which is the extra-respiratory symptom the most prevalent in this setting. “Physical” frailty, considered to be an aging phenotype, has defining traits that can also be considered when studying impaired functional status, but little is known about this relationship in advanced COPD. This review discusses the relevance of this type of frailty in advanced COPD and evaluates it utility and its clinical applicability as a potential outcome measure in palliative care for COPD. *Materials and Methods*: A conceptual review on the functional status as an outcome measure of mortality and morbidity in COPD, and an update on the definition and traits of frailty. *Results*: Data on the prognostic role of frailty in COPD are rather limited, but individual data on traits of frailty demonstrating their relationship with mortality and morbidity in advanced COPD are available and supportive. *Conclusions*: Frailty assessment in COPD patients is becoming a relevant issue not only for its potential prognostic value for increased morbidity or for mortality, but also for its potential role as a measure of functional status in palliative care for advanced COPD.

## 1. Introduction

Chronic obstructive pulmonary disease (COPD) is a debilitating disease that most commonly develops in smokers and is characterized by progressive respiratory symptoms such as dyspnea or coughing. Such symptoms significantly impair the ability to exercise and perform daily activities, and consequently lower the quality of life. In such patients, assessment of the functional status is necessary not only in the more advanced stages of the disease, but also in the earlier stages, because the identification of changes in daily activities can improve therapeutic planning and predict when palliative care might be needed for patients with a more severe form of the disease. “Functional status” is a term that includes a large range of daily physical activities, and often focuses on certain groups rather than an overall quantification. Unfortunately, in COPD, there is no standardized validated approach for evaluating functional status. This can be explained, in part, by the heterogeneity of methods being used in studies on functional status.

Frailty is currently seen as an aging-related phenotype that encompasses several major traits, such as “shrinking” and low activity or weakness [1]. At first glance, many of the defining traits of the frail phenotype are also those used to characterize an impaired functional status. Frailty is an outcome measure broadly used in geriatrics, but it might also have specific utility regarding chronic diseases of the elderly. In COPD patients, especially those in advanced stages of the disease, “frailty” might characterize impairment of the functional status in a more comprehensive manner or serve as a tool to better do-cument the need for palliative care in such patients. This review conceptualizes frailty in COPD patients and advocates for its consideration as an attribute of impaired functional status when the need for palliative care should be considered, regardless of age.

## 2. Functional Status and COPD

Functional status is an outcome measure commonly used to assess the prognosis of various chronic conditions or cancer. It refers to the ability to perform various daily activities, ranging from the basic (e.g., eating and grooming) to the instrumental (e.g., driving and shopping). Initially, functional status was designed to quantify age-related disabilities among the elderly, with or without chronic disease, so that assistance plans could be developed [2]. Subsequently, it was demonstrated that functional status is significantly influenced by a person’s symptom burden, irrespective of disease or age, and this led to a re-evaluation of the relationship between this measure and a patient’s prognosis. This relationship was initially demonstrated in cancer patients, in whom a rapid decline in functional status under curative therapy was found to be associated with a poor prognosis and the need for palliative care [3].

Functional status is used to describe the trajectory in many chronic diseases or in cancer. For example, in COPD and in other organ-specific chronic diseases, functional status worsens with a decline in lung function and an increase in symptom severity. During severe disease exacerbations, functional status undergoes a steep, transitory deterioration followed by incomplete recovery. A functional status-based disease trajectory is commonly associated with palliative care, especially in oncology, whereas in chronic diseases, such as COPD, its importance is rather underestimated. 

In COPD, functional status can be measured by objective standards, such as “field performance” tests (e.g., exercise and handgrip strength), or subjective tools, like a questionnaire, which measures the functional status as described by the patient. In a recent meta-analysis, exercise tests were shown to be the most reliable of the field tests; however, the functional performance inventory questionnaire (original and short form) and the COPD activity rating scale outperformed the objective tests because they offered a more comprehensive view on the deterioration of daily activities [4].

That functional status can be considered a marker of disease severity is not only obvious from the disease trajectory analysis, but also from data coming from various clinical studies on COPD patients: dyspnea severity was found to be inversely correlated with functional status and to be a major limiting factor to physical activity. Among extra-respiratory symptoms, fatigue was found to be associated with impaired exercise capacity in patients with more advanced COPD, irrespective of age or sex [5,6,7,8].

More recently, the prognostic value of functional status in this condition was evaluated in a study performed on stable COPD patients, in which a poor functional status assessed with LCADL was found to be associated with an increased risk of subsequent exacerbations [9]. In another study, functional status was evaluated with the Groningen activity for daily living scale and was shown to be inversely proportional to the risk of death [10].

Functional status has so far been commonly used as an outcome measure of effectiveness in COPD pulmonary rehabilitation programs [11].

The fact that physical performance tests are used interchangeably to measure impairments in functional status, as well as the physical activity traits that define frailty, further supports the consideration of “physical” frailty as a measure of severely impaired functional status for those in the end stage of chronic diseases before palliative care.

## 3. Frailty Phenotype: Dimensions and Domains

Frailty is an emerging unmet medical need, irrespective of chronic disease, because of its complexity and because there are no coherent approaches to limit its impact on daily functioning.

Frailty is a consequence of aging manifested at various levels and associated with an increased risk of falls, dementia, frequent hospitalizations, or mortality. It be defined by meeting at least three of the criteria of the five criteria system (Box 1), whereas pre-frailty requires no more than two [1,12]. These criteria are actually components of the so-called frailty phenotype, classically described by Fried et al. based on the predictors identified in a cohort derived from the U.S. Cardiovascular Health Study, which included over 5200 subjects [1,13]. However, this classical frailty phenotype can be considered a monotrait phenotype when referring to physical frailty [12].

Box 1Defining criteria for physical phenotype [1].Frailty if ≥3 criteria are fulfilledPre-frailty if ≤2 criteria are fulfilled
Fatigue (exhaustion)WeaknessWeight loss (unintentional ≥5 kg over the last 12 months, or sarcopenia)Low speed gaitLow physical activity

Physical frailty can be evaluated using several objective methods such as gait speed, hand grip, standing up, or impedance (body composition). Among these, gait speed and hand grip are the most commonly used; however, impairment in only one of them does not imply impairment in the other, so neither, in isolation, could characterize the patient as frail.

Various studies, though, have evaluated the ability of these variables to predict mortality in chronic conditions. For example, a meta-analysis of 19 studies (12 examining hand grip and 7 gait speed) consistently found that each of these outcome measures predicted a mortality risk independently of other disease-specific factors in patients with cardiovascular disease [14].

The concept of frailty subsequently evolved into a multi-domain phenotype that comprises cognitive, social, economic, and emotional factors. This complex frailty phenotype is currently best investigated using frailty scales such as the Tilburg frailty indicator [15].

This review focused on physical (mono-domain) frailty in an attempt to establish its relationship with functional status and to outline the importance of its identification in routine practice, especially in older COPD patients.

Although frailty is currently screened in older people, its defining criteria can also manifest before the age of 65. In such subjects, pre-frailty is recognized, provided that two defining criteria are detected; if frailty criteria are met, age-related constraints would not be considered probable. In such persons, frailty is probably “masked” by various attributes of disability.

The prevalence of frailty is comparable in both Europe and North America, and shows an increasing trend as a result of increasing life spans. In the frailty phenotype cohort, the calculated overall prevalence was 6.9%, which was shown to increase with age—from 3.2% (65–70) to 25.7% (85–89) [1,13]. In the same study, frailty was associated with a mortality rate of 18% after 3 years and 43% after 7. These rates were higher than those found in pre-frail subjects (7 and 23%, respectively) or in robust adults (3 and 12%, respectively). Hospitalization rates followed the same trend: frail people (59 and 96%, respectively), pre-frail people (43% and 83%, respectively), and healthy subjects (33 and 79%, respectively).

In a subsequent cohort including very old subjects aged at least 76 years old (*n* = 854 patients), the 4-year mortality risk of patients being frail at baseline was 2.7 times higher than that of patients being robust at baseline [16].

In a more recent study in 10 European countries, an analysis of a cohort of 7510 community-dwelling elderly people showed an overall prevalence of 17%: the lowest prevalence was detected in Switzerland (5.8%) and the highest in Spain (27%). The southern countries had a higher prevalence than northern ones [17].

Most of the studies that have evaluated frailty focused on its effects on the community-dwelling elderly; fewer studies were conducted with institutionalized patients, probably based on the fact that such patients are under more constant monitoring and are hardly influenced by frailty. However, several studies were performed with smaller “local” cohorts. One such example is the FINAL study performed at two nursing homes in Spain. It evaluated the prevalence of frailty and its ability to predict mortality and basic daily activities in a sample of predominantly female patients (65.2%) with a mean age of 84.2. The prevalence of frailty at the time was 69.3%. The 1-year mortality rate was 21.6% in frail patients and 17% in robust subjects. During the follow up period, 46.8% of the frail patients developed significant worsening (disability) in their ability to carry out basic daily activities compared with 21.9% of the non-frail patients. Frail patients had a 3.3-times higher risk of mortality or disability compared with non-frail patients, irrespective of age, sex, comorbidities, or other variables examined [18].

The prevalence of frailty was calculated in COPD patients (*n* = 816) undergoing pulmonary rehabilitation programs and was found to be 25.6%. In the same study, this prevalence increased with age, disease severity, and severity of dyspnea [19].

## 4. Frailty and COPD Prognosis

Unlike cardiovascular disease, the relationship between physical frailty and COPD outcomes has only been partially studied. This can be explained in part by a lack of knowledge about the importance of frailty in chronic respiratory disease, and in part by the fact that individual traits, not the phenotype itself, have been evaluated in many studies involving COPD patients.

More recently, attempts to evaluate the relationship between impaired lung function and frailty and their combined effect on mortality risk were made in the cohort of the Cardiovascular Health Study (*n* = 3578). However, this cohort included a mixed population with chronic obstruction and chronic restriction, not only COPD patients. Frailty status was measured over a period of 3 years, the development of lung function impairment was monitored over a period of 4 years, and the death rate over a median follow-up of 13.2 years. Baseline pre-frailty and frailty rates were 48.3 and 5.8%, respectively, and were associated with an increased risk of airflow limitation (probability ratio of 1.62:1 for pre-frailty and 1.88:1 for frailty). Frail people without lung function impairment were significantly more likely to develop it compared with robust persons (probability 1.42:1). The most interesting finding of this study was that patients with impaired lung function had an increased risk of developing frailty features (probability 1.58:1)

The association of frailty with impaired lung function was found to result in the highest risk of death (3.91:1) compared with those who were not frail or did not have any impairment of their lung function [20].

In the Rotterdam Study cohort of 2142 participants, frailty was found to have a prevalence of 10.2% in the subset of subjects with COPD compared with 3.4% of the subset without COPD (*p* < 0.001). The probability of frailty was 2.2 times higher in these patients (*p* = 0.002) irrespective of age, sex, smoking, or corticosteroid use. The prevalence of frailty was found to increase with airflow limitation, dyspnea severity, and the number of previous exacerbations [21].

In another study involving COPD outpatients (*n* = 121 patients), the prevalence of frailty was evaluated using a FRAIL scale (fatigue, resistance, ambulation, chronic illness, and weight loss). The prevalence of frailty was 6.6% and pre-frailty 41.3%, and fatigue was the most commonly detected criterion in the studied population. The predictors of frailty were found to be comorbid cancer, at least two hospital admissions over the previous 12 months, obesity, and the presence of sarcopenia [22].

The presence of COPD was also found to increase the risk of developing frailty with an increased risk of progressing from reversible to irreversible: this was demonstrated by an analysis carried out on 5086 men, 8% being frail and 46% being pre-frail. Over a median follow up period of 4.6 years, 35% of the pre-frail or robust males progressed to frailty or died. Among the predictors of this pattern were limitations in functional status, higher IL-6 levels, and the presence of COPD or diabetes mellitus [23].

Studies that evaluated the relationship between individual frailty traits and COPD mortality or morbidity did not consider them to be measures of frailty but measures of a prognosis. However, the information from such studies is important because it provides “indirect” supportive data for the importance of pre-frail or frail states in the progression and burden of this disease. Table 1 summarizes the studies that evaluated these traits in COPD patients. The data from these studies are also discussed in more detail below.

## 5. Muscle Weakness and Deconditioning and COPD Prognosis

Hand grip and sit-to-stand tests are validated outcome measures for mortality in the general population, as well as in subsets of chronic diseases, where they are most commonly used for their predictive value in patients with heart disease [14].

In a large population study performed in 17 countries with 139,691 patients enrolled, the evaluation of hand grip strength for all causes attributable to COPD or asthma (mortality, cardiovascular mortality, mortality, and morbidity) was among the main outcomes evaluated over 4 years. The all-cause mortality rate was 2%. Grip strength was identified as a predictor of all-cause mortality, with a 5 kg reduction in grip strength being associated with a hazard ratio for mortality of 1.16:1 (*p* < 0.0001), and for both cardiovascular and non-cardiovascular mortality of 1.17:1 (*p* < 0.0001). Grip strength was not identified as a predictor of hospital admission for COPD exacerbations [30].

The Obstructive Lung Disease in Northern Sweden (OLIN) study was a large-cohort study that examined the impact of various clinical and functional variables on disease outcomes: hand grip was assessed in 441 COPD patients and 570 non-COPD patients as an outcome measure for impaired muscle strength. Even though the hand grip assessment did not show any significant differences between males and females (25.8 kg versus 26.9 kg, *p* = 0.05 in females; 45.9 kg versus 46.3 kg *p* = 0.6 in males), it did show a significant impairment in more advanced COPD (FEV1%pred < 50%; females 21.4 kg versus 26.9, *p* = 0.010; males 41.5 kg versus 46.3 kg, *p* = 0.038). The presence of comorbid heart disease was associated with a more impaired hand grip in both males and females with COPD, irrespective of the severity of airway obstruction (females 22.9 versus 26.3, *p* = 0.005; males 46.9 versus 41.9 *p* = 0.003) [31]. This study did not examine the prognostic value of hand grip impairment in the COPD population or if its impairment was associated with more frequent exacerbations.

In the Korean National Health and Nutrition Examination Survey, which comprised 14,930 subjects, a cohort of 832 COPD patients was identified and matched with an equal number of non-COPD subjects. Similarly, hand grip was not found to differ between the two cohorts, except for males (38 versus 38.9 kg, *p* = 0.04), in whom hand grip was also found to correlate with quality of life [24]. In this study, quality of life was evaluated using a generic questionnaire and no relationship with mortality or morbidity was analyzed for the hand grip in this cohort.

In another study performed on a smaller sample of patients (*n* = 88) with advanced COPD (FEV1%pred 34.2 ± 15.2%), impaired hand grip was associated with worse health [32].

Impaired physical performance is common in COPD, but it can be difficult to assess in routine clinical practice. Timed up-and-go (TUG) test and other easily applied assessments of physical performance were compared with the 6-min walk distance (6MWD). In a longitudinal study of comorbidities in COPD, submaximal physical performance was determined in 520 patients and 150 controls using the TUG test and 6MWD. Spirometry, body composition, handgrip strength, the COPD assessment test, St George’s Respiratory Questionnaire (SGRQ), and the modified Medical Research Council dyspnea scale were also used. The patients and controls were similar in age, body mass index, and sex proportions. The TUG in the patients was greater than that in the control group (*p* = 0.001), was inversely related to the 6MWD (*r* = −0.71, *p* < 0.001), and forced one-second predicted expiratory volume (*r* = −0.19, *p* < 0.01). It was directly related to the SGRQ activity (*r* = 0.39, *p* < 0.001), SGRQ total (*r* = 0.37, *p* < 0.001), and total COPD assessment test scores (*r* = 0.37, *p* < 0.001). The TUG test identified the difference in physical performance between patients and controls, and together with the validated questionnaires provided a rapid measure of physical performance that could be used in clinical practice [25].

The relationship among the hand grip and sit-to-stand tests, health status, and morbidity and mortality in COPD patients was analyzed in a prospective study on a cohort of 409 stable COPD patients enrolled at primary care level in Switzerland and The Netherlands. This cohort included 57% males, most of whom (63.8%) were COPD patients with less advanced diseases (FEV1%predicted > 50%). The mortality rate over a 2-year observation period was 9.3%. Sit-to-stand test performance was poorer in females, in patients with advanced COPD, and in patients with more severe dyspnea or more comorbidities. The hand grip test was more effective among females and in patients with more severe dyspnea, and did not seem to be influenced by the severity of airflow limitation [33]. Both the sit-to-stand and hand grip tests were found to predict mortality, but not subsequent exacerbations, in this cohort (see Table 1) [33].

A similar analysis was performed on the cohort during a 5-year follow up: the mortality rate was found to be 19% and the sit-to-stand test was again identified as a predictor of mortality: each extra three-time repetition was associated with a reduction in mortality risk of 0.81, whereas impaired hand grip was not significantly associated with an increase in mortality likelihood. Neither of the tests could predict exacerbations [10].

## 6. Fatigue and COPD Prognosis

Fatigue can be generally ascribed to weakness and was found to be the most frequently diagnosed extra-pulmonary symptom in COPD, as well as one of the therapeutic targets for pulmonary rehabilitation [34].

It can be measured in COPD patients by using various scales or domains of health-related quality-of-life questionnaires, such as the Chronic Respiratory Disease Questionnaire [34].

Fatigue was found to increase in severity with the progression of the disease and to significantly affect health [26], and was documented as a predictor for mortality in the OLIN cohort, where this relationship was analyzed in 434 patients. Fatigue was evaluated with a version of the FACIT-F questionnaire and was found to predict mortality (OR = 1.06) and a poor quality of life [27].

## 7. Weight Loss/Sarcopenia and COPD Prognosis

When malnutrition was assessed as a part of a nutritional evaluation using the ESPEN criteria to diagnose malnutrition in COPD patients, weight loss was found to be associated with an increased risk of mortality (probability 2.72:1 *p* = 0.08 over 6 months, and *p* = 0.06 over 9 months) in a cohort of patients hospitalized for a COPD exacerbation [35]. Malnutrition assessed with the same criteria set was also found to predict 2-year COPD-related mortality and morbidity (hospital admissions and their severity, expressed as the duration of hospital stays). In the analyzed cohort, the prevalence of malnutrition was found to be 24.6%, and its presence was associated with a four-fold increase in mortality risk and a 3.55:1 probability of an increase in the risk of subsequent hospitalization [28].

As expected, extreme nutritional deficit (cachexia) was also shown to be an independent predictor of mortality in a large cohort of COPD patients (HR 2.16, *p* < 0.001) Sarcopenia is a comorbid state in patients with COPD. Its severity has been found to increase with age and physical deconditioning. In a study involving 622 patients, sarcopenia was detected in 14.5%, and was found to increase in frequency in older, more severe COPD patients. COPD patients with sarcopenia had worse functional and health status [36].

## 8. Exercise Capacity and COPD Prognosis

Exercise capacity can be used as an outcome measure of poor gait speed or impaired physical activity. Some exercise capacity tests measure the distance walked in meters (6-min walking tests or shuttle tests, for example) and this variable can be used as a prognostic endpoint or as an endpoint of therapeutic efficacy in rehabilitation programs alone or combined with certain pharmacological interventions.

Similarly, nutritional status and exercise capacity were also shown to be predictors of mortality or morbidity in COPD patients, independently or when included in the BODE index [37,38,39]. In the ECLIPSE study cohort (2010 COPD patients), for example, 6-min walked distance, 6-min walked speed, 6-min walking work, distance-saturation product, and exercise-induced oxygen desaturation were identified as predictors of 3-year mortality (probability 2.30:1, 2.15:1, 2.17:1, 2.7:1, and 1.75:1, respectively), whereas 6-min walking work was, among others, found to predict hospitalizations for COPD exacerbations (1.23:1) [38].

## 9. Frailty, Frailty Traits, and Their Value as Markers of Disease Severity and Their Clinical Applicability

From the data discussed above, it emerged that frailty, as a measure of impaired functional status, is very useful for documenting the severity of COPD.

## 10. Corollary: Frailty in Advanced COPD and Its Potential Role as an Outcome Measure in Palliative Care

The previous studies discussed above clearly delineated a relationship between frailty and an increased risk of morbidity or mortality. In COPD, frailty is scarcely studied, even though data are being made available to define the criteria for frailty (e.g., sarcopenia, weight loss, or fatigue) rather than frailty itself.

In other chronic progressive diseases such as heart disease, frailty was found to be associated with a more adverse prognosis. In COPD, such a relationship has not yet been investigated, but it might be of particular interest to elderly patients who belong to the frequent exacerbator phenotype. This COPD phenotype has attracted particular interest over the last few years because of its association with increased mortality and increased use of healthcare resources. Therefore, attempts to document what makes such patients more prone to exacerbations—severe exacerbations in particular—have identified various factors related to etiology (exposure to smoking), disease pathogenesis (more prominent bronchial inflammation), pathogenic epiphenomena (chronic airway infection), nutritional status, and comorbidities. Physical deconditioning is also thought to play a role in this disease outcome, but little data are available to strongly support this.

Most importantly, frailty in itself is still unrecognized in COPD, despite its demonstrated relevance in other populations of patients. In particular, in COPD, the predictive capacity of frailty for mortality will be of particular interest in relation to elderly patients in subsequent years, and it might be both interesting and useful to study frailty as a potential predictor of frequent hospitalizations for elderly patients who experience frequent exacerbations. Such supportive data are not only useful for research purposes, but are also important for the routine care of frail COPD patients, who might benefit from a more integrated, multidisciplinary approach that targets major unmet medical needs, including frailty.

The relevance of frailty in COPD is not only based on the fact that many patients with this disease develop it after 65 years of age, but also because both the state (frailty) and the disease (COPD) negatively influence each other: frailty can increase the morbidity and mortality risk, whereas COPD can accelerate the progression of a pre-frail state to frailty or aggravate the latter. Identifying COPD patients at risk of becoming frail can trigger therapeutic measures to reverse pre-frailty or improve frailty, and thus have the potential to improve the outcome of the underlying disease indirectly.

The identification of frailty, especially so-called “irreversible” frailty, in COPD patients is particularly important in advanced COPD, because this triggers an assessment for palliative care. As functional status is a key outcome measure in palliative care and because frailty, based on the data discussed above, can be considered a measure of severely impaired functional status, it is now clear that frailty should be considered in COPD when this type of care is required and should be taken into account when evaluating the prognosis of the disease.

To conclude, frailty assessment in COPD patients has become a relevant issue not only for its potential prognostic value regarding increased morbidity or mortality, but also for its potential role as a measure of functional status in palliative care for advanced COPD patients.

## Figures and Tables

**Table 1 medicina-57-00474-t001:** Summary of the studies analyzing defining criteria of physical frailty as prognostic factors for mortality and morbidity in chronic obstructive pulmonary disease (COPD).

Outcome Measure for Frailty Defining Criteria	Sample Size	Predictor of Mortality	Predictor of Morbidity (Future Exacerbations)
Hand grip strength test [24]	139,691 (various pathologies including COPD)	Yes: HR = 1.17 (*p* < 0.0001) for non-cardiovascular mortality	No
Hand grip strength test [25]	409	Yes: HR = 0.84 (*p* = 0.04)	No
Sit to stand test [25]	409	Yes: HR = 0.58 (*p* = 0.004)	No
Fatigue [26]	434	Yes: OR = 1.06	Not measured
Malnutrition [27]	121	Borderline OR 2.72 (*p* = 0.06)	Not measured
Malnutrition [28]	118	Yes: HR 3.9 (*p* = 0.009)	HR = 3.55 (*p* = 0.042)
Cachexia [28]	1755	Yes HR = 2.16, *p* < 0.001	No
6MWT distance [29]	2010	YES: OR 2.30	No

## Data Availability

Not applicable.

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
