# Peer review of "Frailty, a Dimension of Impaired Functional Status in Advanced COPD: Utility and Clinical Applicability"

_medicina, 2021, doi:10.3390/medicina57050474_

Round 1

Reviewer 1 Report

The authors are addressing the very important topic of Frailty in COPD.  This has been somewhat under appreciated over the years.

Major comments:

The Manuscript needs extensive editing for grammar and syntax.  

Lines 42-52 and 52-61 are identical.

Some of the measures included ( such as fatigue or malnutrition) are not always frailty, and others ( such as he BODE index ) include other metrics, such as lung function.  The authors have to be careful not to confuse the reader.

According to Table 1- The BODE index is protective for mortality ( this is not true).

Author Response

Major comments:

The Manuscript needs extensive editing for grammar and syntax.  Thank you. This was done in house using MDPI author service.

Lines 42-52 and 52-61 are identical. We are kindly asking the Editorial Office to highlight this for us so that we then ca remove the redundant info

Some of the measures included (such as fatigue or malnutrition) are not always frailty, and others ( such as he BODE index ) include other metrics, such as lung function.  The authors have to be careful not to confuse the reader. Thank you for your very valuable comments. Indeed fatigue or malnutrition are not always frailty but this was a discussion (an analogy) done in order to strengthen the prognostic value of physical frailty in general and of component features in particular.   

According to Table 1- The BODE index is protective for mortality ( this is not true). Thank you very much. The data was wrongly inserted in the table. And the inclusion of the BODE index might be indeed confusing so that this information was excluded in both draft and Table.

Reviewer 2 Report

I appreciate this original review, covering a topic which has not been extensively investigated.

I have two comments:

  • A discussion chapter about the overall studies should be included
  • Please have a deep english review. I've seen several errors throughout the paper. 

Author Response

I appreciate this original review, covering a topic which has not been extensively investigated.

I have two comments:

  • A discussion chapter about the overall studies should be included. Thank you indeed this discussion was necessary and the authors inserted a new section  which should better introduce the last section on the frailty and copd palliative care.
  • Please have a deep english review. I've seen several errors throughout the paper. This was done in house using MDPI Author Services.  

Round 2

Reviewer 1 Report

The version made available to me did not have tables.  These needed changes and I need to see them prior to completing review

Author Response

Dear Reviewer

On behalf of the authors I appologize for this shortcoming. I did not notice that the version returned from the English check had no tables included. For your consideration I attach them to this correspondence and I will also send them to the Editorial Office.

Thank you in advance, any comment which may help to improve the quality of this draft is very much appreciated

Sabina

Reviewer 2 Report

For me now it's ok. 

Author Response

On behalf of the authors I do thank you for your comments.

Sincerely,

Sabina

Round 3

Reviewer 1 Report

OK- This has been responsive to my concerns and can warrants publication.